# Increased pupal temperature has reversible effects on thermal performance and irreversible effects on immune system and fecundity in adult ladybirds

David N. Awde[1,2], Michal Řeřicha[1] & Michal Knapp [ID] [1✉]

The environmental conditions an organism encounters during development vary in their lasting impact on adult phenotypes. In the context of ongoing climate change, it is particularly relevant to understand how high developmental temperatures can impact adult traits, and whether these effects persist or diminish during adulthood. Here, we assessed the effects of pupal temperature (17 °C – normal temperature, 26 °C – increased temperature, or 35 °C – heat wave) on adult *Harmonia axyridis* thermal stress tolerance, immune function, starvation resistance, and fecundity. The temperature during pupation significantly affected all investigated traits in fresh adults. Heat acclimation decreased adult haemocyte concentration, cold tolerance, and total egg production, and had a positive effect on heat tolerance and starvation resistance. The negative effects of heat acclimation on cold tolerance diminished after seven days. In contrast, heat acclimation had a lasting positive effect on adult heat tolerance. Our results provide a broad assessment of the effects of developmental thermal acclimation on *H. axyridis* adult phenotypes. The relative plasticity of several adult traits after thermal acclimation may be consequential for the future geographic distribution and local performance of various insect species.

[1] Department of Ecology, Faculty of Environmental Sciences, Czech University of Life Sciences Prague, Kamýcká 129, 165 00 Prague - Suchdol, Czech Republic. [2] Department of Biology, Faculty of Science, Mount Saint Vincent University, Halifax, NS, Canada. ✉email: knapp@fzp.czu.cz

Performance, survival, and fitness of ectotherms are heavily influenced by seasonal and annual variations in temperature[1,2]. This is especially important considering the increasing variability in global and regional climates and the increased frequency of extreme weather events (e.g., heat waves and droughts) caused by ongoing climate change[3]. It is imperative to describe the consequences of these shifts in abiotic conditions on organismal performance at a species level to effectively pre-serve and protect biodiversity in agricultural and natural ecosystems[4]. Animals respond to environmental changes across two timelines. Across generations, selection can act on heritable traits, which can increase the fitness of select individuals experiencing environmental change[5]. Within generations, an individual's phenotype depends on the interaction between their genotype and the environmental conditions they are exposed to[6,7]. Therefore, animal traits are partially dependent on previous exposure(s) to thermal conditions[8,9]. When considering the thermal environment, thermal acclimation refers to a phenotypic change in response to long-term exposure to a thermal condition (days to months; summarized in[10]).

In insects, a thermal acclimation response can depend on the timing of the exposure since different developmental stages have different capacities for thermal acclimation[11,12]. Thermal acclimation as an adult appears to be more easily reversible than acclimation during juvenile development[13,14]. Because of this, thermal acclimation during juvenile stages (developmental acclimation) is often considered irreversible (see[15,16]). However, while morphological changes caused by developmental thermal acclimation are likely irreversible[14], physiological and behavioural changes remain relatively plastic[17]. For example, the thermal limits of adult *Drosophila melanogaster* are influenced by the thermal environment experienced as larvae, and this effect can be reversed after eclosion with adult thermal acclimation[17]. Degut et al.[13] showed that developmental temperature irreversibly influences the morphology of adult *Pieris napi* (body mass, wing shape, and size), which may drive the impact of developmental temperature on adult behavioural and physiological traits such as flight distance, flight endurance, and reproductive output. Whether other seemingly unrelated adult traits, such as immune function and starvation resistance remain plastic after developmental thermal acclimation, and how those traits may interact or correlate in their response remains largely unclear.

To investigate this phenomenon, we performed laboratory experiments using the harlequin ladybird, *Harmonia axyridis*, as model insect species. This ladybird is one of the most invasive species worldwide[18], resulting in good, however, yet incomplete, knowledge of its thermal biology (e.g.[19–23]; and reviewed with other coccinellids by Sloggett[24]). In general, ladybirds exposed to colder temperatures lay fewer eggs, live longer, and have higher haemocyte concentrations than ladybirds exposed to warm conditions[20]. With respect to the effects of thermal acclimation on adult phenotypes, *H. axyridis* appears to have varied thermal plasticity depending on the timing of thermal exposure and whether that exposure is hot or cold[23,25]. Adult acclimation to above average temperatures does improve heat tolerance, but acclimation as larvae do not elicit any change in adult heat tolerance[15]. Exposure to cold shocks before a cold-temperature challenge improves the cold tolerance of adults[26], but prolonged exposure to cold does not improve cold tolerance[25].

During winter temperatures, haemocyte concentrations are reduced in adults[27], and there is also a marginal reduction for humoral components of the immune system (e.g., antimicrobial activity against *Escherichia coli*[28]), but the effect of developmental thermal acclimation on the immune system is unknown. In contrast, exposure to prolonged cold temperatures during development results in higher antioxidative enzyme activity in

freshly eclosed adults than exposure to high temperatures, but this effect fades after 24 h[29]. These results suggest that thermal acclimation may lead ladybirds to invest in self-maintenance mechanisms (at least temporarily), and this reallocation of resources potentially occurs at the expense of future reproduction[30]. However, it is unclear how acclimation to high temperatures, specifically when ladybirds must endure thermal stress while immobile (as pupae), may improve or negatively affect important adult phenotypes.

In this study, we assessed adult thermal tolerance (chill comma recovery time and heat knockdown time) and immune system functioning (haemocyte concentration) of adult *H. axyridis* after experiencing one of three thermal environments as pupae. These thermal environments represented an approximate standard daily temperature (17 °C – normal temperature), an above average temperature (26 °C – increased temperature), and a temperature akin to a heat wave (35 °C) during the ladybird active-season in Central Europe (April – September; Czech Hydrometeorological Institute). Moreover, after eclosion, we maintained all ladybirds at a consistent temperature (the median of all three treatments; 26 °C) and described the lasting or diminishing effects of each acclimation treatment by comparing the traits of 1- and 7-day old adults. We also compared the reproductive capacity and starvation resistance of treated ladybirds. Conclusions drawn from this study provide a more accurate framework of the lasting effects of developmental thermal acclimation in insects and provide a thorough assessment of the potential effects of climate change, specifically hot temperatures, on insect performance and changing geographic distributions.

## Results

**Thermal performance**. Pupal temperature had a dramatic effect on CCRt shortly after eclosion; ladybirds acclimated to 17 °C as pupae had significantly shorter CCRt than ladybirds reared at 26 °C and 35 °C (Fig. 1; Table S1; Temperature main effect: $p < 0.001$). However, differences between pupal temperatures disappeared in adult ladybirds assayed 7 days post-eclosion (Fig. 1; Table S1; Temperature*Day interaction effect: $p < 0.001$). Interestingly, CCRt for ladybirds exposed to 26 °C and 35 °C as pupae were reduced during the first 7 days of adult life and reached levels similar to ladybirds exposed to 17 °C. There was no effect of ladybird size or sex on chill coma recovery time (Table S1; Live mass main effect: $p = 0.83$; Sex main effect: $p = 0.39$).

Compared to cold tolerance (CCRt), HKDt values were similar for adult ladybirds measured 1 and 7 days after eclosion (Fig. 1; Table S1; Day main effect: $p = 0.30$; Temperature*Day interaction effect $p = 0.46$). However, freshly eclosed ladybirds that experienced 35 °C as pupae had significantly longer HKDt than those that experienced 26 °C (Fig. 1; Table S1; Temperature main effect: $p = 0.001$). Interestingly, this difference was not statistically significant in pairwise comparisons for ladybirds measured 7 days after eclosion (Fig. 1). There was no effect of ladybird dry mass or sex on HKDt (Table S1; Dry mass main effect: $p = 0.80$; Sex main effect: $p = 0.17$).

**Haemocyte concentration**. Pupal temperature and adult age significantly affected haemocyte concentration (Table S1; Temperature main effect: $p < 0.001$; Day main effect: $p < 0.001$). Significant differences were observed among all pupal temperatures for freshly eclosed adults (Fig. 2), haemocyte concentration being negatively related to pupal temperature. The same pattern was observed in 7-day-old ladybirds; however, statistically significant differences were observed only between ladybirds exposed to 17 °C and 35 °C (Fig. 2). Haemocyte concentration significantly

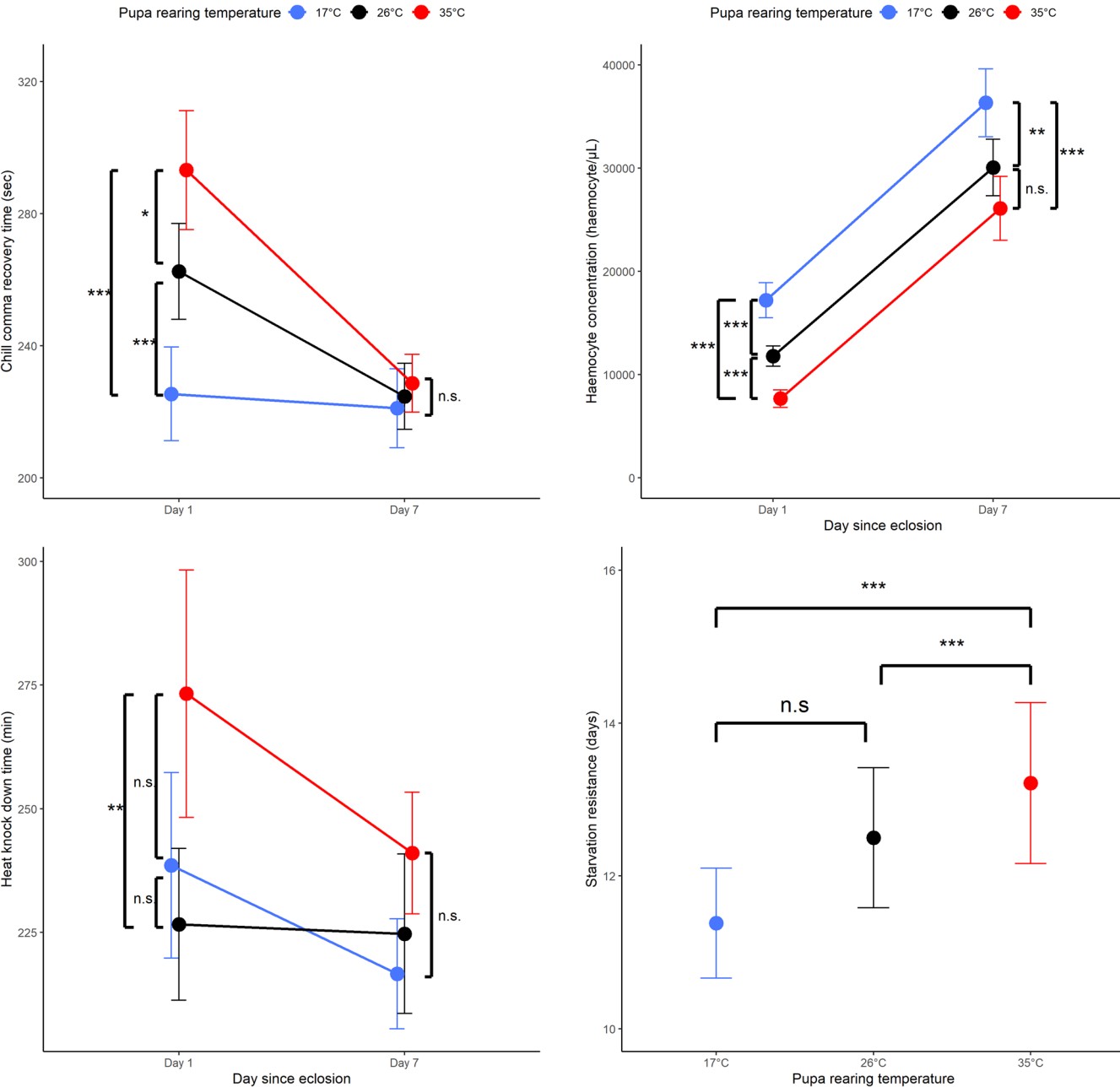

**Fig. 1 Effects of acclimation temperature and day since exposure on chill coma recovery time (CCRt) and heat tolerance (HKDt).** Adult *Harmonia axyridis* recovery times in response to cold exposure (top) and knockdown times in response to heat exposure (bottom), after acclimation to one of three temperatures as pupa (red = 35 °C; black = 26 °C; blue = 17 °C). Means are plotted ± 95% confidence intervals (*denote pairwise comparisons in which $p < 0.05$, **$p < 0.01$, and ***$p < 0.001$).

**Fig. 2 Effects of acclimation temperature and day since exposure on haemocyte concentration and starvation resistance.** Adult *Harmonia axyridis* haemocyte concentrations (top) and starvation resistance (bottom) after acclimation to one of three temperatures as pupa (red = 35 °C; black = 26 °C; blue = 17 °C). Means are plotted ± 95% confidence intervals (*$p < 0.05$, **$p < 0.01$, and ***$p < 0.001$).

increased during early adult life, and there was also a significant effect of sex, with males having higher haemocyte concentrations than females (Figure S1; Table S1; Sex main effect: $p < 0.001$).

**Starvation resistance**. Pupal temperature had a significant effect on adult starvation resistance (Table S1; Temperature main effect: $p < 0.001$). Ladybirds reared at 35 °C during the pupal stage lived significantly longer than ladybirds exposed to 26 °C and 17 °C (Fig. 2). Ladybird size and sex also significantly influenced longevity without food, with large ladybirds living longer than small ones and males living longer than females (Table S1; Fig. S2;

Fig. S3; Live mass main effect: $p < 0.001$; Sex main effect: $p < 0.001$).

**Fecundity**. Pupal temperature strongly influenced the cumulative egg production across 28 days (Fig. 3; Table S1; Temperature*-Day interaction effect: $p = 0.02$). Females exposed to 35 °C during the pupal stage laid fewer eggs than those exposed to 26 °C and 17 °C. We found no size effect on cumulative egg production (Table S1; Fig. S3; Live mass main effect: $p = 0.34$) or pre-oviposition period (Table S1). In contrast to cumulative egg production, developmental temperature had no significant effect

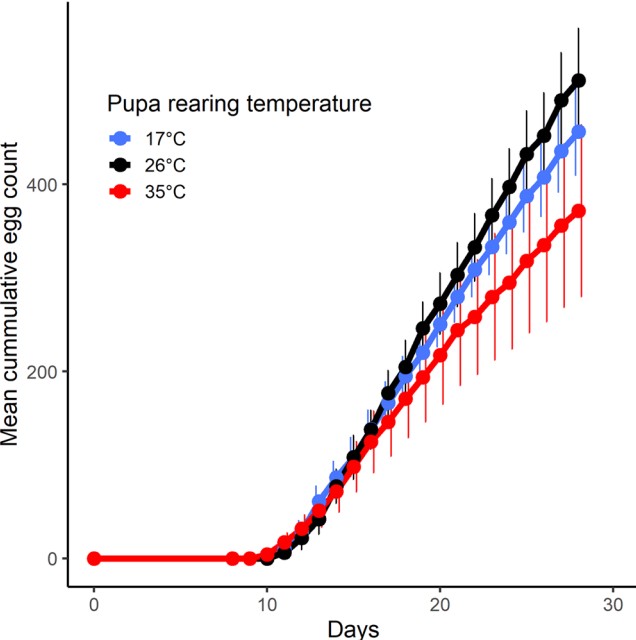

**Fig. 3 Effects of developmental temperature on *Harmonia axyridis* fecundity.** Mean ± 95% confidence intervals of cumulative egg production per treatment (red = 35 °C; black = 26 °C; blue = 17 °C) per female.

on the preoviposition period (Table S1; Temperature main effect: $p = 0.23$).

## Discussion

Here, we show that increasing developmental temperatures impose a cost on the performance of some adult ladybird traits, however, high temperature and starvation tolerance improved with heat acclimation. Thermal acclimation during pupation varied in its initial and lasting effects on adult thermal tolerance performance and had a lasting effect on immune function. Therefore, we successfully demonstrated the wide variation in adult trait plasticity possible after the development of thermal acclimation. Together, our results indicate that the reversible or irreversible nature of thermal acclimation is largely influenced by the relative plasticity of a specific trait. Below, we discuss the effects of developmental thermal acclimation on phenotypes assayed in this study and the implications of this information on our understanding of insect performance in the context of ongoing climate change.

The developmental temperature had a lasting effect on adult immune function, and heat tolerance, and sufficiently influenced the physiology of adult ladybirds so that starvation resistance and fecundity differed between treatment groups. Specifically, 35 °C acclimation significantly reduced haemocyte concentrations compared to cooler treatments, and this effect was still present after seven days. The 35 °C treatment is akin to recent and predicted heat waves[31,32], and high temperatures during development have been shown to decrease haemocyte concentrations in another insect, the moth *Lobesia botrana*[33]. In adult insects, thermal stress primes immune responses[34], possibly because of cross-protection of activated stress-related molecules such as protective cellular chaperones (i.e., heat shock proteins[34–36]); Therefore, and considering our results, it is possible that thermal stress enhances the immune response immediately (previous studies[34–36]), while exposure to thermal stress during development may impose a cost on the immune function of adults.

The whole-body effects of developmental thermal acclimation on ladybirds led to a somewhat paradoxical result with respect to starvation resistance and fecundity metrics. On one hand, acclimation to extreme heat reduced cumulative egg production, consistent with previous work[37]. On the other hand, the same acclimation treatment increased starvation resistance when compared to standard or above average conditions, as is the case in *D. melanogaster* and a butterfly species, *Bicyclus anynana*[38,39]. A possible explanation for these results is that acclimation to extreme heat during development resulted in an investment in self-maintenance at the expense of reproduction[30]. This survival-reproduction trade-off has been observed in *D. melanogaster* after repeated stressful exposures to cold[40,41]. In insects, increased heat tolerance and longevity are frequently positively correlated[42], while fecundity decreases with increased heat tolerance[43]. Increased starvation resistance may result from heat acclimation-related changes in metabolism[44] and the expression of heat shock proteins[45]. Female reproductive failure would not be caused solely by female-specific acclimation to extreme heat, but decreased fecundity could be related to high temperature effects during pupation that indirectly impacts sperm production and quality in adults[46], as heat wave-exposed females were mated by heat wave-exposed males in our experiment. The ability to mitigate (increase starvation resistance and heat tolerance), and not necessarily excel (reduced fecundity and hemocyte counts), when faced with heat stress may be one contributing factor in the rapid range expansion of *H. axyridis*.

Cold tolerance remained plastic in adults after developmental thermal acclimation (at least seven days post-eclosion). Interestingly, the negative consequences of acclimation to extreme heat on cold tolerance were lost after seven days, while the effects of acclimation to the normal temperature on cold tolerance did not differ between 1-day and 7-day old ladybirds. In ladybirds, as in other insects, cold and heat tolerance varies with age[17,26,47]. Ladybirds acclimated to normal and the increased temperature had a high level of cold tolerance performance 1 and 7 days after eclosion, which was not the case for ladybirds exposed to heat waves during the pupal stage. This may be particularly relevant in the context of climate change and consistently increasing temperatures[3]. Annually, insects are exposed to warmer pupal temperatures during late summer (potential heat acclimation), leaving newly eclosed adults especially vulnerable to rapid cold snaps (i.e., late summer nights). However, since adults maintain cold tolerance plasticity after developmental thermal acclimation, the effects of warming pupal temperatures likely pose less of a threat to cold stress tolerance, at least in comparison to the other traits measured in this study.

The effects of developmental heat acclimation on heat tolerance did not fully diminish with age, showing limited adult plasticity. The difference in relative plasticity between cold and heat tolerance is consistent with previous studies indicating that basal performance of these two traits rarely correlates in other insect species, and these traits differ in their underlying genetic and physiological mechanisms[48–50]. Moreover, as in *D. melanogaster*[17], ladybird cold tolerance remained plastic in adults, while adult heat tolerance appears more constrained by development thermal acclimation, suggesting that these phenotypic characteristics may be generalizable across species[17,51,52]. Therefore, we hypothesize that the fixed effects of developmental thermal acclimation on adult heat tolerance will be consequential for many insects faced with increasing thermal variability (but see[8]). For example, ladybirds exposed to lower temperatures as pupae in spring may be unable to cope with or further acclimate to rising summer temperatures and heat waves.

In conclusion, our data clearly demonstrate the variation in adult plasticity after developmental thermal acclimation across important adult traits such as stress tolerance and fecundity. We provide data that supports the hypothesis that in the context of

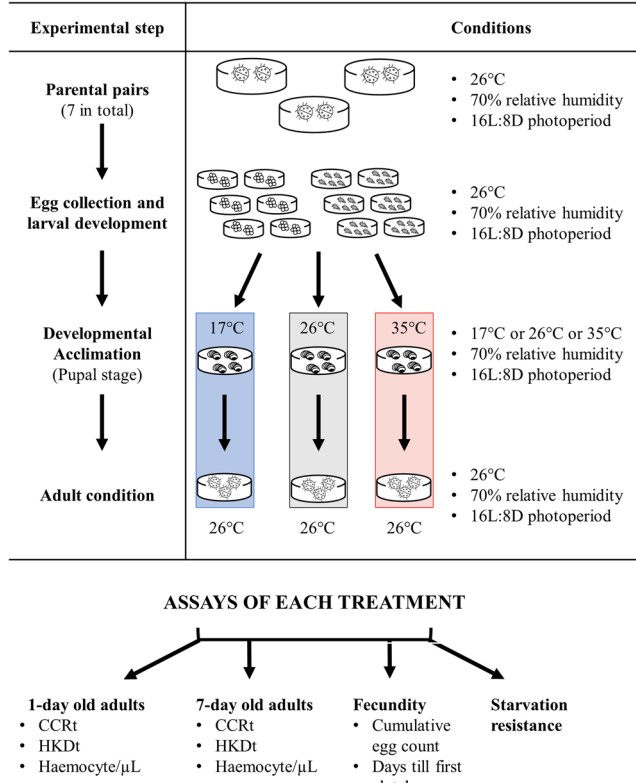

**Fig. 4 *Harmonia axyridis* rearing conditions and experimental set up.** Graphic representation of the environmental conditions of parental and experimental generations of *H. axyridis* at each specific life stage.

changing climate, it may be more important to consider developmental conditions influencing adult heat tolerance than adult cold tolerance since the latter appears better able to adjust to immediate thermal conditions. Finally, we show that increasing temperatures during an immobile developmental stage imposes a lasting cost on important adult traits (fecundity and immune function), which may affect the performance of insects exposed to ongoing climate change.

## Methods

**Ladybird rearing and experimental setup.** Individuals assayed in our experiment were the offspring of field-collected adults from České Budějovice, Czech Republic (48.9794344 N, 14.4451989 E). Seven parental pairs were placed into separate Petri dishes (9 cm in diameter) containing folded paper strips as a substrate for egg laying (Fig. 4). Parental pairs were kept in a climatic chamber (Sanyo® MIR-155) at 26 °C, relative humidity of 70%, and long day photoperiod (16L:8D). Ladybirds were provided with fresh pea aphids and water *ad libitum* daily. Eggs from each parental pair were collected, each clutch placed into separate Petri dishes, and kept under the same conditions as parental pairs. After hatching, larvae were separated into smaller groups of 4–5 individuals, placed into new Petri dishes, and provided with fresh pea aphids and water *ad libitum*. The last instar larvae were placed individually into new Petri dishes, fed until the prepupal stage, and then checked twice a day to collect fresh pupae as soon as possible. Petri dishes with fresh pupae were immediately distributed among three thermal chambers set to one of three thermal regimes: normal temperature (17 °C), increased temperature (26 °C), or heat wave (35 °C; Fig. 4). Pupae were checked twice a day (morning and evening) to collect freshly eclosed adults. The pupal stage was selected because of its lack of mobility, and therefore its inability to escape such high stressful temperatures under natural conditions.

New adults were moved to the 26 °C climatic chamber and provided with pea aphids and water *ad libitum* (Fig. 4). After 24 h, the first subset of adults was used to measure haemocyte concentration and thermal performance. Seven days after eclosion, the same variables were measured for the second subset of adults. The third subset of ladybirds was used to assay ladybird longevity without food. Finally, the fourth subset was used to assay fecundity (e.g., preoviposition period, number of clutches, etc.).

**Thermal performance.** To investigate the high- and low-temperature performance of ladybirds subjected to various pupal temperatures, we measured chill coma recovery time (CCRt[53,54]); and heat knockdown time (HKDt[55]); To measure CCRt, sets of six ladybirds in separate Petri dishes were exposed to −4 °C in a climatic chamber for 120 min and then transferred to a climatic chamber set to 26 °C for recovering. Beetles were turned onto their dorsum and time until the first coordinated movement was measured (e.g., flipping to their ventral side). Experimental ladybirds were stimulated to move by gentle teasing with an entomological pin every fifteen seconds to identify beetles that remained motionless despite being already able to move. To measure HKDt, sets of 12 individuals (in a few cases less) were placed in clean Petri dishes (without food and water) in a climatic chamber set to 42 °C. Ladybirds were checked every 15 min using a webcam placed inside the chamber. Individuals that had become immobile or dead (reached their HKDt) were easily identified as motionless, typically with spread wings. We aimed to analyse up to six individuals for each combination of variables (sex, parental pair, pupal temperature, and sampling time; *n* = 336 ladybirds). However, some parental pairs produced an insufficient number of offspring, resulting in 289 beetles measured for CCRT and 270 for HKDT (raw data available in the Dryad digital repository[56]). As a measure of body size, we recorded the live body mass after the eclosion of beetles assayed for CCRt using a Sartorius balance with a precision of $10^{-4}$ g. All beetles measured for HKDt were dried for 48 h at 42 °C after the HKDt assay and their dry mass was recorded.

**Haemocyte concentration.** Haemolymph was collected via puncture sampling[57]. Haemolymph was collected using a glass microcapillary (Hirschmann, Germany) and measured using a digital calliper with a precision of 0.01 mm. Collected haemolymph (~1 µl) was immediately diluted (100× dilution) in anticoagulant buffer (Phosphate-buffered saline: 137 mM NaCl, 2.7 mM KCl, 10 mM Na2HPO4, and 1.8 mM KH2PO4), and the total haemocyte concentration was recorded immediately using a Bürker chamber under a Carl Zeiss Primo Star microscope (set to 100× magnification). We aimed to analyze four individuals for each combination of sex, parental pair, pupal temperature, and sampling time (1 day or 7 days; *n* = 336 ladybirds). However, problems sampling several individuals (e.g., haemolymph coagulation) resulted in a smaller dataset of 308 beetles[56].

**Starvation resistance.** We indirectly tracked the whole-body effects of developmental thermal acclimation on ladybird physiology by assessing longevity under a physiologically stressful condition, food deprivation. Ladybirds used to investigate the effects of pupal temperature on starvation resistance (longevity without food) were placed in Petri dishes with watered pieces of cotton wool and moved to a climatic chamber set to 26 °C. Survival of these ladybirds was checked daily, and new water was added *ad libitum*. In total, starvation resistance was recorded for 126 individuals, i.e., three individuals per combination of sex (2), parental pair (7), and pupal temperature (3). Their live body mass after eclosion was measured using a Sartorius balance with a precision of $10^{-4}$ g since body mass is known to influence starvation resistance in ladybirds[58].

**Fecundity.** Finally, we evaluated the fitness consequences of thermal acclimation during pupation by tracking the cumulative egg production and days until the first clutch of females after eclosion. We recorded egg production each of 28 days after eclosion. Experimental pairs (*n* = 63 males and females) were established from a combination of individuals that eclosed on the same day and originated from different parental pairs. Each experimental pair was placed in one Petri dish for mating, placed in a climatic chamber set to 26 °C, relative humidity of 70%, and long day photoperiod (16L:8D; same as the parental generation), and provided with pea aphids and water *ad libitum*. Every day folded paper strips in Petri dishes were changed and the total number of eggs was counted. Note that ladybirds typically lay eggs on the underside of leaves, which is mimicked by folded paper strips. Live body mass was measured for each experimental female after eclosion.

**Statistics and reproducibility.** Ladybird responses to developmental thermal acclimation were evaluated using generalized linear mixed models (GLMMs). Specifically, we evaluated the response variables CCRt, HKDt, haemocyte concentration, starvation resistance, cumulative egg production, and days until the first clutch using six separate GLMMs with appropriate error distributions (see the next paragraph) and a combination of predictor variables (depending on the availability) in the order of size, sex, acclimation temperature (treated as a categorical variable), and day since eclosion (Table S2). Models were selected to address the biological question of this study, which was to evaluate the effects of pupal acclimation temperature, time since exposure, and the interaction between temperature and time on each response variable. We included size and sex factors in the model to understand how these two variables may also influence each response variable. To account for any genetic effects on the response variables, we included the parental pair (Family) of each ladybird as a random factor in each of these models. We compared the cumulative egg production of acclimated females with a slightly different model construction compared to the other five response variables (Table S2). We treated the day of each observation and the ID of each ladybird as random factors (Day|ID), allowing both the slope and intercept to vary, which

accounted for repeated sampling of each individual female over time and account for potential ontogenetic changes in egg production over time.

All statistics and figures were performed and created in R and RStudio, version 4.2.1 and 2022.02.3, respectively. We used the function *check_distribution* from the package *performance* to assess the probability that a given variable belongs to a specific distribution (Table S2). We used the *glmmTMB* function from the *glmmTMB* package to analyse CCRt, HKDt, haemocyte concentration, starvation resistance, cumulative egg production, and preoviposition period. The performance of each model was visually inspected using the *check_model* function from the *performance* package. We generated summary statistics of each model using the *Anova (type "III")* function from the *Car* package. Pairwise comparisons were performed with Tukey's tests using the *lsmeans* function from the *lsmeans* package. Figures were created using *ggplot2*.

**Reporting summary**. Further information on research design is available in the Nature Portfolio Reporting Summary linked to this article.

## Data availability

Raw data used to produce the figures and support the findings of this study are available in the Dryad Digital Repository https://doi.org/10.5061/dryad.1c59zw413.

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

## Acknowledgements

We are grateful to Ezequiel González for his detailed comments on the draft of our manuscript. We thank Radek Svoboda for help with the laboratory experiment and Oldřich Nedvěd for sharing his knowledge and laboratory at the University of South Bohemia. This study was supported by the Ministry of Education, Youth and Sports of the Czech Republic (project number CZ.02.2.69/0.0/0.0/18_053/0016979) and partly by the OP VVV project: CZ.02.2.69/0.0/0.0/19_073/0016944 (students grant no. 71/2021).

## Author contributions

All authors contributed (DNA, MŘ, and MK) contributed equally to the production of this manuscript.

## Competing interests

The authors declare no competing interests.
