## [Peer Review File · Communications Biology]

Reviewers' comments:

Reviewer #1 (Remarks to the Author):

This study investigates the effects of temperature during the pupal stage on various traits during adult stage by putting all adults back to the same temperature. This way, authors were able to determine which traits display lasting effects of pupal acclimation temperature versus traits that demonstrate plasticity by changing across adult age. I find the question of interest, and the experimental design is well-conceived. The paper reads very well.

My first main comment is that the direct effect of temperature treatment on the developmental time of pupae is not indicated. This may be needed to partition between the indirect effect on adult traits of pupae temperature-induced change in adult body size and the direct effect of pupae acclimation on adult performance. Although not mandatory, this data may be a nice addition.

My second main comment is that at several places the authors conclusions appear a bit speculative relative to the results shown here. This is linked to the attempt to wrap the results into the context of the insect response to climate change. I recommend the authors to be careful in their statements at these places:

Line 269: there was no immune challenge imposed on adults here, therefore I do not understand why the authors state that their result contrasts with previous studies?

Line 288: do we know if in this species the males elaborate their sperm during the pupal stage? or adult stage? Because again, here in this experiment the adults were not exposed to 'heat wave' temperature'.

Line 299: do they have data to demonstrate that pupae are actually experiencing increasing temperatures in their microhabitat (supposedly plant surfaces or soil)? The shortcut here is a bit easy. Moreover, 35°C constant during several days (time needed for pupae to emerge), does NOT reflect a heat wave event which by definition reach a peak temperature during the day but we still have temperature fluctuation daily (not constant).

Line 314: this study does not explore this facet because adults were not exposed to challenging temperatures (that was not the objective).

Minor points:

Line 185: ...acclimation temperATURE...

Line 194: it is first time I see the use of GLMM with time as random factor to run a repeated measure analysis - I suspect this makes the GLMM essentially similar to a RM-ANOVA, but the authors may want to provide support to this approach (reference). Alternatively, why not applying a RM-ANOVA?

Reviewer #3 (Remarks to the Author):

General comments

Temperature is one of the most important abiotic factors for the performance of insects. How insects respond to thermal stress has become a major concern in a warming world in recent decades. The study examined the effects of three pupal temperatures (17 °C, 26 °C, and 35 °C) on thermal tolerance, haemocyte concentration, starvation resistance, and fecundity in *Harmonia axyridis*. The authors found that increased pupal temperature significantly reduced cold tolerance, and had a potential to enhance heat tolerance in 1-day-old ladybeetles, and probably reduced haemocyte concentration in 1- and 7-day-old ladybeetles. Pupae exposed to 35 °C markedly increased starvation resistance and decreased fecundity. Generally, the experiments were well designed, and the manuscript was well organized. However, some points are worth discussing and the manuscript could be further improved.

Specific comments

TITLE

I would like to use 'pupal temperature' instead of 'developmental temperature'. The latter could be included as a keyword.

ABSTRACT

L17-19: Long sentence. Please rephrase it.

L27: seven days, or 7 days. However, Arabic numbers less than ten are always spelled out. Please check the guidelines for authors.

KEYWORDS

Delete 'Insects'

INTRODUCTION

L50-51: The references 11-12 do not suitable for this sentence. They do not refer to acclimation, i.e., exposure to a thermal condition for days or months. You may find articles published in *Entomologia Generalis*, *Journal of Pest Science*, and *Pest Management Science* to support this sentence.

L88-95: 17 °C is defined as a normal temperature, but adults were maintained at 26 °C, an increased temperature. Why not maintain them at 17 °C? It seems more reasonable to maintain at the normal temperature.

MATERIALS AND METHODS

L134: A maximum of 12 individuals were used, what is the maximum value? Please directly show it in the main text.

L136: It's interesting that a webcam was used here. I wonder if it is clear enough to see the ladybeetle, which is small. Could you please show us some pictures? It is better to show important parameters of the webcam in the text.

L138-142: It is hard to follow. It confuses me. Sets of six individuals were used to measure CCRt, but the total number was 289, so how many replicates were there? It was supposed to use four individuals for each combination, but a total of 289 individuals were tested, so how many individuals were there for each combination? You should rephrase these sentences so that readers can clearly understand the experimental design without having to refer to dataset files.

L142-144: Why was live body mass tested in the CCRt assay while dry body mass was recorded in the HKDt assay?

L164-166: see comment on L138-142

L174: add information on relative humidity and photoperiod.

L198: 2022.02.3, respectively.

RESULTS

I strongly recommend listing Table S2 in the main text. However, the table does not look good, and should be reorganized to make it 'beautiful' and easy to read. If you follow my suggestions, the main

text of this part should be rephrased.

L224: Use 'haemocyte concentration' instead.

L246-247: What is 'time until first clutch'? Do you mean the preoviposition period?

DISCUSSION

According to the sequence in the RESULTS section, it is better to discuss in the order: cold tolerance, heat tolerance, haemocyte concentration, starvation resistance, and fecundity.

Line 252-256: First, those two sentences do not 'connect' well. Second, the second sentence is not easy to follow.

L263: There were no lasting effects on heat tolerance. Please check it.

L270-271: 'possibly because ofheat shock proteins', this is not clearly enough. Please provide more details.

L271-274: The sentence is not easy to follow. In addition, what do you mean 'impose a cost on the long-term immune function of adults'? Do you consider 7 days as a long term?

L289-290: What do you think it is about the ability of the ladybeetle to mitigate the influence of heat stress? I can't find any evidence in this paragraph.

L312-314: I don't think you can draw such a conclusion from the present study.

REFERENCES

Almost all Latin names are not italicized.

L410: Check the journal name.

FIGURES

Figure 3: HAEMOCYTE CONCENTRATION: $p < 0.001$ between 17 and 26, 26 and 35. But $p < 0.01$ between 17 and 35? Not reasonable.

Figure 3: Starvation resistance: I do not think there is significant difference between 26 and 35. Because there is much overlap between the values, even if they are shown in mean \pm SE. Please check it. In addition, I want to know how you check the validation of model fitting.

Figure 4: The 95% confidence intervals for each point are missing.

SUPPLEMENTARY MATERIALS

Supplementary Table S2: First, show it in the main text. Second, reorganize it to make it look good. Third, how do you get these data (like anova table)? You did not mention it in the text.

Replace 'Supplementary Figure 1-3' with 'Supplementary Figure S1-3'.

Reviewer #4 (Remarks to the Author):

The current work evaluated the impact of increase temperature to assess the effect of pupal temperature on adult ladybirds. They looked at different aspects : immune function, starvation resistance, fecundity, chilli recovery time and heat knockdown time. The authors found interesting results, like heat acclimation decreased adult haemocyte concentration, cold tolerance and total egg production. But they found a positive effect on starvation resistance.

The current paper represent lot of works with lot of individuals used for the experiment.

Beside the scientific quality of the current work, the starvation experiment as well as the heat knockdown experiment could have been avoided and the added value is not worth to the unethical consideration of individuals. Of course, this is a personal point of view.

Specific comments on the paper

Abstract,

The abstract is not very easy to understand solely. Of course after reading the paper it's easier.

The authors could clarify "in their initial and lasting impact"

The second sentence "in the context of ongoing climate change". Could the authors precise why their experiment is relevant in this context? I'll have also a comment on this part in the introduction.

L41 and 42 of the introduction, could the authors be more specific on what they meant regarding the understanding consequences of shifts in abiotic conditions and the protection of the biodiversity?

Especially did they have any solution? What is the potential application of the current study in this context?

The authors used the word "acclimation" but do not always specify if it's temperature /thermal acclimation or not. (L51; L58) and some times "developmental acclimation. May be the authors could standardize the terminology to facilitate the reading.

L65. Not sure that it's really an "issue"

L66 Ladybird is presented as a vey invasive species, but without context. For year the goal was to introduced it and they were artificially released in huge quantity in the environment. As it a voluntary release for years, it surprising to introduce this point in the context of the current study. Could the authors precise what is the link between the current study and the fact the species is considerer as invasive?

Regarding the M et M, it's clearly explained.

The authors could specify which volume of Haemocyte they collected per individual to be able to do further analysis. Were the individuals alive after collection or how were they killed?

L165- 166 What is the link with live body mass and the starvation resistance. Could the authors explained it? This information could may be mentioned in another section?

The results are clearly explained and the figures well presented.

Regarding the discussion, could the authors explain, what their study add in the context of climate change, as this point was highlighted in their introduction?

Reviewers' comments:

Reviewer #1 (Remarks to the Author):

This study investigates the effects of temperature during the pupal stage on various traits during adult stage by putting all adults back to the same temperature. This way, authors were able to determine which traits display lasting effects of pupal acclimation temperature versus traits that demonstrate plasticity by changing across adult age. I find the question of interest, and the experimental design is well-conceived. The paper reads very well.

Dear reviewer,

Thank you for your positive evaluation and constructive comments!

My first main comment is that the direct effect of temperature treatment on the developmental time of pupae is not indicated. This may be needed to partition between the indirect effect on adult traits of pupae temperature-induced change in adult body size and the direct effect of pupae acclimation on adult performance. Although not mandatory, this data may be a nice addition.

Only one analysis showed an effect of adult body size on a response variable – starvation resistance.

- For the remaining phenotypes we assayed, there was no effect of body size. This indicates that the indirect effects of pupal temperature (i.e., by altering adult body size) did not have a large effect on each of these phenotypes.
- For starvation resistance, we did find that pupation temperature significantly affected adult live mass ($p < 0.001$), with ladybirds acclimated to the hottest condition being smaller than the other two temperatures. This does indicate a direct effect of temperature on an adult trait which could indirectly affect starvation resistance, however, we do not think this is the case because of our results. We found that large ladybirds were the most resistant to starvation, and that ladybirds acclimated to the hot condition were more resistant to starvation than those acclimated to cooler conditions. If the indirect effects of pupal temperature had a major effect on starvation resistance, then we would predict that ladybirds exposed to cold pupal temperatures should be more starvation resistant because they are larger. This was not the case.

My second main comment is that at several places the authors conclusions appear a bit speculative relative to the results shown here. This is linked to the attempt to wrap the results into the context of the insect response to climate change. I recommend the authors to be careful in their statements at these places:

Line 269: there was no immune challenge imposed on adults here, therefore I do not understand why the authors state that their result contrasts with previous studies?

We have changed this sentence to better reflect the exact result from the cited papers instead of confusing those studies (with an immune challenge) and our own study.

Line 271: “In adult insects, thermal stress primes immune responses³⁹, possibly because of cross-protection of activated stress-related molecules such as protective cellular chaperones (i.e., heat shock proteins;³⁹⁻⁴¹). Therefore, and considering our results, it is possible that thermal stress enhances the immune response immediately (previous studies³⁹⁻⁴¹), while exposure to thermal stress during development may impose a cost on the immune function of adults.”

Line 288: do we know if in this species the males elaborate their sperm during the pupal stage? or adult stage? Because again, here in this experiment the adults were not exposed to 'heat wave' temperature'.

We have changed this section of the text to better indicate that we are proposing that potential developmental effects from temperature may have downstream consequences on reproductive physiology in adults, and not that heat effects during pupation directly effect sperm production during development. We have edited the sentence to better reflect this.

Line 288 “Female reproductive failure would not be caused solely by female-specific acclimation to extreme heat, but decreased fecundity could be related to high temperature effects during pupation that indirectly impact sperm production and quality in adults⁵¹, as heat wave-exposed females were mated by heat wave-exposed males in our experiment.”

Line 299: do they have data to demonstrate that pupae are actually experiencing increasing temperatures in their microhabitat (supposedly plant surfaces or soil)? The shortcut here is a bit easy. Moreover, 35°C constant during several days (time needed for pupae to emerge), does NOT reflect a heat wave event which by definition reach a peak temperature during the day but we still have temperature fluctuation daily (not constant).

We do not have data to say exactly what temperatures pupae experience in the field. However, we do think it is reasonable to suggest that pupae likely experience high temperatures in their micro habitats (plant surface) when ambient temperatures are high since the microhabitat in which pupation occurs does not change throughout the breeding season (similar plant surfaces or artificial structures, e.g., walls, from spring to autumn; personal observations).

To address the second point, we do recognize that temperatures in the field fluctuate daily, but as with many thermal tolerance studies, we simplified our testing to stable temperatures. Note that heatwaves in Central Europe can represent average temperature of 35°C (in shade) over several consecutive days (30°C at night and 40°C during daytime).

Line 314: this study does not explore this facet because adults were not exposed to challenging temperatures (that was not the objective).

We have changed this line to indicate that this is a prediction we generated based on our results.

Line: 317 “For example, ladybirds exposed to lower temperatures as pupae in spring may be unable to cope with or further acclimate to rising summer temperatures and heat waves.”

Minor points:

Line 185: ...acclimation temperATURE...

We have corrected this spelling error.

Line 194: it is first time I see the use of GLMM with time as random factor to run a repeated measure analysis - I suspect this makes the GLMM essentially similar to a RM-ANOVA, but the authors may want to provide support to this approach (reference). Alternatively, why not applying a RM-ANOVA?

This comment is correct, the model is functioning in a way that is similar to a RM-ANOVA. However, we opted to use the GLMM because our data (cumulative egg count) was not normally distributed (negative binomial distribution) and was not suitable for basic RM-ANOVA approach.

Reviewer #3 (Remarks to the Author):

General comments

Temperature is one of the most important abiotic factors for the performance of insects. How insects respond to thermal stress has become a major concern in a warming world in recent decades. The study examined the effects of three pupal temperatures (17 °C, 26 °C, and 35 °C) on thermal tolerance, haemocyte concentration, starvation resistance, and fecundity in *Harmonia axyridis*. The authors found that increased pupal temperature significantly reduced cold tolerance, and had a potential to enhance heat tolerance in 1-day-old ladybeetles, and probably reduced haemocyte concentration in 1- and 7-day-old ladybeetles. Pupae exposed to 35 °C markedly increased starvation resistance and decreased fecundity. Generally, the experiments were well designed, and the manuscript was well organized. However, some points are worth discussing and the manuscript could be further improved.

Dear reviewer,

Thank you for your positive evaluation and constructive comments!

Specific comments

TITLE

I would like to use 'pupal temperature' instead of 'developmental temperature'. The latter could be included as a keyword.

We have made this change to the title.

ABSTRACT

L17-19: Long sentence. Please rephrase it.

We have edited this sentence.

Line 17: “The environmental conditions an organism encounters during development vary in their lasting impact on adult phenotypes.”

L27: seven days, or 7 days. However, Arabic numbers less than ten are always spelled out. Please check the guidelines for authors.

We have made this change. We could not find any mention in the guidelines, but we do agree in this case and similar errors in the remaining text. All of those errors have been corrected.

KEYWORDS

Delete ‘Insects’

Deleted.

INTRODUCTION

L50-51: The references 11-12 do not suitable for this sentence. They do not refer to acclimation, i.e., exposure to a thermal condition for days or months. You may find articles published in *Entomologia Generalis*, *Journal of Pest Science*, and *Pest Management Science* to support this sentence.

We have fixed this mistake and have used two references that are more appropriate.

Line 48: “In insects, a thermal acclimation response can depend on the timing of the exposure since different developmental stages have different capacities for thermal acclimation^{11,12}.”

11. Mutamiswa, R., Machezano, H., Chidawanyika, F., & Nyamukondiwa C. Life-stage related responses to combined effects of acclimation temperature and humidity on the thermal tolerance of *Chilo partellus* (Sweinhoe) (Lepidoptera: Crambidae). *J. Therm. Biol.* 75, 85–94 (2019).

12. Gray, E. M. Thermal acclimation in a complex life cycle: the effects of larval and adult thermal conditions on metabolic rate and heat resistance in *Culex pipiens* (Diptera: Culicidae). *J. Insect. Physiol.* 59, 1001–1007 (2013).

L88-95: 17 °C is defined as a normal temperature, but adults were maintained at 26 °C, an increased temperature. Why not maintain them at 17 °C? It seems more reasonable to maintain at the normal temperature.

Although 17 °C is a normal temperature throughout the active season in the field in the Czech Republic, temperatures used for rearing adult ladybirds, facilitating mating, and oviposition in the lab are often higher (e.g., 20°C and 25°C in Knapp and Nedvěd 2013; 26 °C in Knapp and Řeřicha 2020). We made the experimental decision in this study to use the increased

temperature (26°C) as our rearing temperature and post-acclimation temperature because it is the median of the three treatments.

MATERIALS AND METHODS

L134: A maximum of 12 individuals were used, what is the maximum value? Please directly show it in the main text.

We have changed the sentence to be clear. We used maximally 12 individuals at a time to measure HKDt (per batch).

Line 133: “To measure HKDt, sets of maximally 12 individuals were placed in clean Petri dishes (without food and water) in a climatic chamber set to 42°C.”

L136: It's interesting that a webcam was used here. I wonder if it is clear enough to see the ladybeetle, which is small. Could you please show us some pictures? It is better to show important parameters of the webcam in the text.

We have used webcams to measure heat tolerance for smaller insects like *Drosophila* (see Awde et al 2020 *JoVE* for video). The webcam offers an advantage over direct observation when monitoring insects since we can remove the observer from the insect sight and keep the animal in a very stable environment that is fully enclosed (incubator). Further, we can maintain videos for future use (not in this study). Attached is a sample photograph from a recent heat knockdown assay in our lab (unpublished) using ladybirds and a similar protocol.

L138-142: It is hard to follow. It confuses me. Sets of six individuals were used to measure CCRt, but the total number was 289, so how many replicates were there? It was supposed to use four individuals for each combination, but a total of 289 individuals were tested, so how many individuals were there for each combination? You should rephrase these sentences so that readers can clearly understand the experimental design without having to refer to dataset files.

We have edited the sentence to indicate that *up to six individuals* were used per parental pair. We disagree that providing the exact number of individuals per combination (7 parental pairs x 2 sexes x 3 temperature treatments x 2 sampling days x 2 thermal tolerance measurements) will provide meaningful additional information that is not already easily accessed from the raw data (simple sorting or pivot chart in excel).

Line 137: “We aimed to analyse up to six individuals for each combination of variables (sex, parental pair, pupal temperature, and sampling time; n = 336 ladybirds). However, some

parental pairs produced an insufficient number of offspring, resulting in 289 beetles measured for CCRT and 270 for HKDT (REF to the Dryad file containing raw data).”

L142-144: Why was live body mass tested in the CCRT assay while dry body mass was recorded in the HKDT assay?

The difference in size measurements is an artifact from previous plans to assess the effects of heat stress on ladybird wet weight. These plans did not pan out and we did not pursue these analyses. However, we do not expect any differences between outputs of analyses employing dry vs live body masses.

L164-166: see comment on L138-142

We have edited this sentence to include the number of factors in each variable to make it easier to understand how the total number (n=126) was generated.

Line 163: “In total, starvation resistance was recorded for 126 individuals, i.e., three individuals per combination of sex (2), parental pair (7), and pupal temperature (3).”

L174: add information on relative humidity and photoperiod.

We have added this information.

Line 173: “Each experimental pair was placed in one Petri dish for mating, placed in a climatic chamber set to 26°C, relative humidity of 70%, and long day photoperiod (16L:8D; same as the parental generation), and provided with pea aphids and water *ad libitum*.”

L198: 2022.02.3, respectively.

We have corrected this error

RESULTS

I strongly recommend listing Table S2 in the main text. However, the table does not look good, and should be reorganized to make it ‘beautiful’ and easy to read. If you follow my suggestions, the main text of this part should be rephrased.

We politely disagree with this suggestion. The summary statistics from each model (important to understand the main patterns in the data) are provide in the main text and figures. We do not think that including a table that takes up an entire page would be a good use of space in the main text. As for the suggestion to make the table ‘beautiful’, we think the table is formatted to best display the results of our GLMMs in a way that is easy to survey each response variable efficiently (we do not have ideas how to improve it).

L224: Use ‘haemocyte concentration’ instead.

We have made this change.

L246-247: What is ‘time until first clutch’? Do you mean the preoviposition period?

Yes, that is what we meant, and we have changed the wording. We also changed the wording in the relevant tables.

Line 250: “In contrast to cumulative egg production, developmental temperature had no significant effect on preoviposition period (Table S2; Temperature main effect: $p = 0.23$).”

DISCUSSION

According to the sequence in the RESULTS section, it is better to discuss in the order: cold tolerance, heat tolerance, haemocyte concentration, starvation resistance, and fecundity.

We have made a stylistic choice to discuss our results in the order we think is the most interesting and important to readers. Furthermore, we think it is effective to discuss the response variables that showed similar plastic responses to acclimation together.

Line 252-256: First, those two sentences do not ‘connect’ well. Second, the second sentence is not easy to follow.

We have deleted the first sentence and re worded the second sentence to make it easier to follow.

Line: 255: “Here, we show that increasing developmental temperatures impose a cost the performance of some adult ladybird traits, however, high temperature and starvation tolerance improved with heat acclimation.”

L263: There were no lasting effects on heat tolerance. Please check it.

From the GLMM, there was an overall lasting effect on heat tolerance since we did not find a statistically significant effect of day on heat knock down time (HKDt). Below is the specific section from the results.

Line 218: “Compared to cold tolerance (CCRt), HKDt values were similar for adult ladybirds measured 1 and 7 days after eclosion (Figure 2; Table S2; Day main effect: $p = 0.30$; Temperature*Day interaction effect $p = 0.46$).”

We think this justifies the line in question (Line 263 in the discussion).

L270-271: ‘possibly because ofheat shock proteins’, this is not clearly enough. Please provide more details.

We have added details to indicate the potential function involved in the protective effect that these proteins may have. We think that adding more details, or elaborating further will be too speculative and we are much more comfortable putting forward our results to support an already existing hypothesis.

Line 271: “In adult insects, thermal stress primes immune responses³⁹, possibly because of cross-protection of activated stress-related molecules such as protective cellular chaperones (i.e., heat shock proteins;³⁹⁻⁴¹).”

L271-274: The sentence is not easy to follow. In addition, what do you mean ‘impose a cost on the long-term immune function of adults’? Do you consider 7 days as a long term?

We have edited the sentence to be clearer and clarify that we meant the effects from development to adult stages.

Line 273: “Therefore, and considering our results, it is possible that thermal stress enhances the immune response immediately (previous studies³⁹⁻⁴¹), while exposure to thermal stress during development may impose a cost on the immune function of adults.”

L289-290: What do you think it is about the ability of the ladybeetle to mitigate the influence of heat stress? I can't find any evidence in this paragraph.

We have edited the sentence to directly point to the results we are using to justify these claims. The results are discussed in the paragraph as well.

Line 291: “. The ability to mitigate (increase starvation resistance and heat tolerance), and not necessarily excel (reduced fecundity and hemocyte counts), when faced with heat stress may be one contributing factor in the rapid range expansion of *H. axyridis*.”

L312-314: I don't think you can draw such a conclusion from the present study.

Correct, this sentence conveys how we think acclimation during pupation might impact a range of insect species. We have edited this sentence to clearly indicate that this is a hypothesis we have generated from our study.

Line 315: “Therefore, we hypothesize that the fixed effects of developmental thermal acclimation on adult heat tolerance will be consequential for many insects faced with increasing thermal variability (but see⁸).”

REFERENCES

Almost all Latin names are not italicized.

We have fixed this error.

L410: Check the journal name.

We have fixed this error.

FIGURES

Figure 3: HAEMOCYTE CONCENTRATION: $p < 0.001$ between 17 and 26, 26 and 35. But $p < 0.01$ between 17 and 35? Not reasonable.

Thank you for pointing out this error. We have fixed it, $p < 0.001$ between 17 and 35.

Figure 3: Starvation resistance: I do not think there is significant difference between 26 and 35. Because there is much overlap between the values, even if they are shown in mean \pm SE. Please check it. In addition, I want to know how you check the validation of model fitting.

We have double checked the result and can confirm that based on our statistical analyses that those values are significantly different.

We have added details to the text describing how we validated our models.

Line 202: “The performance of each model was visually inspected using the *check_model* function from the *performance* package.”

Figure 4: The 95% confidence intervals for each point are missing.

For clarity and to remove clutter, we decided to remove the error bars for this figure. Attached is the figure with the error bars.

SUPPLEMENTARY MATERIALS

Supplementary Table S2: First, show it in the main text. Second, reorganize it to make it look good. Third, how do you get these data (like anova table)? You did not mention it in the text.

We politely disagree with this point (mentioned in an earlier comment). In addition, we have added the method we used to get the anova table.

Line 204: “We generated summary statistics of each model using the *Anova* (type “III”) function from the *Car* package.”

Replace ‘Supplementary Figure 1-3’ with ‘Supplementary Figure S1-3’.

We have made this change.

Reviewer #4 (Remarks to the Author):

The current work evaluated the impact of increase temperature to assess the effect of pupal temperature on adult ladybirds. They looked at different aspects : immune function, starvation resistance, fecundity, chilli recovery time and heat knockdown time. The authors found interesting results, like heat acclimation decreased adult haemocyte concentration, cold tolerance and total egg production. But they found a positive effect on starvation resistance.

Dear reviewer,

Thank you for your positive evaluation and constructive comments!

The current paper represent lot of works with lot of individuals used for the experiment.

Beside the scientific quality of the current work, the starvation experiment as well as the heat knockdown experiment could have been avoided and the added value is not worth to the unethical consideration of individuals. Of course, this is a personal point of view.

We appreciate this point of view and will consider it in the future. In this type of research, it is difficult to avoid since we are directly assessing physiological limits. However, in many of our (Insect Ecology Group) field research activities we try to keep ethic considerations in our minds and limit unnecessary killing of insects (by selecting suitable sampling techniques and appropriate experimental designs).

Specific comments on the paper

Abstract,

The abstract is not very easy to understand solely. Of course after reading the paper it's easier.

We have edited the entire abstract to simplify several sentences that describe results from the main text. This has made the abstract much easier to understand.

The authors could clarify “in their initial and lasting impact”

We have edited the sentence in question and have removed this problematic phrase.

Line 17: “The environmental conditions an organism encounters during development vary in their lasting impact on adult phenotypes.”

The second sentence “in the context of ongoing climate change”. Could the authors precise why their experiment is relevant in this context? I'll have also a comment on this part in the introduction.

We feel we have placed our experiments in this context based on the rest of the sentence.

Line 18: “In the context of ongoing climate change, it is particularly relevant to understand how high developmental temperatures can impact adult traits, and whether these effects persist or diminish during adulthood.”

L41 and 42 of the introduction, could the authors be more specific on what they meant regarding the understanding consequences of shifts in abiotic conditions and the protection of the biodiversity? Especially did they have any solution? What is the potential application of the current study in this context?

We have edited the sentence to specifically highlight what we think is most important. The sentence is meant to place our study in a broad context of how species level descriptions are important. We think that trying to provide solutions or the potential applications of unknown results at this stage in the text is not important for a reader. We do provide our take on how our results might suggest how changing climates could impact insects at large.

Line 38: “It is imperative to describe the consequences of these shifts in abiotic conditions on organismal performance at a species level to effectively preserve and protect biodiversity in agricultural and natural ecosystems⁴.”

The authors used the word “acclimation” but do not always specify if it's temperature /thermal acclimation or not. (L51; L58) and some times “developmental acclimation. May be the authors could standardize the terminology to facilitate the reading.

We have standardized throughout the entire text to facilitate easier reading, as suggested by this comment.

L65. Not sure that it's really an "issue"

We have changed the word "issue" to "phenomenon" to better reflect what we are describing in the paragraph.

L66 Ladybird is presented as a very invasive species, but without context. For years the goal was to introduce it and they were artificially released in huge quantity in the environment. As it a voluntary release for years, it surprising to introduce this point in the context of the current study. Could the authors precise what is the link between the current study and the fact the species is considered as invasive?

We do agree with some of this assessment however, the harlequin ladybird is well established as an invasive species in non-native habitats, outside of an agricultural context (see Brown et al 2011). Framing this ladybird species as invasive is consistent with much of the previous ecology and physiological studies on the species and does accurately describe current habitat ranges well (since plenty are outside of an agricultural ecosystem). The difference between surviving an artificial release long enough to perform an agricultural service is very different than being able to establish a population able to survive annually without intervention.

We have added a sentence to the text to better relate the current study to the invasive classification of the species.

Line 94: "Conclusions drawn from this study provide a more accurate framework of the lasting effects of developmental thermal acclimation in insects and provides a thorough assessment of the potential effects of climate change, specifically hot temperatures, on insect performance and changing geographic distributions."

Regarding the M et M, it's clearly explained.

The authors could specify which volume of Haemocyte they collected per individual to be able to do further analysis. Were the individuals alive after collection or how were they killed?

The volume of haemolymph sampled was a single drop, ~ 0.7-1µl. We have added this detail to the text (see below). Ladybird haemolymph can be collected via reflex bleeding (see Knapp et al. 2018, 2020), which does not result in ladybird death, and is a natural response by ladybirds to predators. We are killing experimental animals by fast (deep) freezing.

Line 146: "Haemolymph was collected using a glass microcapillary (Hirschmann, Germany) and measured using a digital calliper with a precision of 0.01 mm. Collected haemolymph (~1µl) was immediately diluted (100× dilution) in anticoagulant buffer (Phosphate-buffered saline: 137 mM NaCl, 2.7 mM KCl, 10 mM Na₂HPO₄, and 1.8 mM KH₂PO₄), and the total haemocyte concentration was recorded immediately using a Bürker chamber under a Carl Zeiss Primo Star microscope (set to 100× magnification)."

L165- 166 What is the link with live body mass and the starvation resistance. Could the authors explained it? This information could may be mentioned in another section?

Body mass has a significant impact on how long a ladybird can survive stressful conditions and without food (Knapp and Řeřicha, 2020).

We have added this detail to the text.

Line 164: “Their live body mass after eclosion was measured using a Sartorius balance with a precision of 10^{-4} g since body mass is known to influence starvation resistance in ladybirds³⁵.”

The results are clearly explained and the figures well presented.

Regarding the discussion, could the authors explain, what their study add in the context of climate change, as this point was highlighted in their introduction?

We feel we have provided this information in the discussion. Below is an example.

Line 315: “Therefore, we hypothesize that the fixed effects of developmental thermal acclimation on adult heat tolerance will be consequential for many insects faced with increasing thermal variability (but see⁸). For example, ladybirds exposed to lower temperatures as pupae in spring may be unable to cope with or further acclimate to rising summer temperatures and heat waves.”

REVIEWERS' COMMENTS:

Reviewer #1 (Remarks to the Author):

I am happy with the changes the authors provided to their MS - I think the paper has improved in impact.

Just one more thing:

Line 255: impose a cost ON the performance

Reviewer #3 (Remarks to the Author):

I agree with the other two reviewers that the ms is well prepared. However, it could be further improved. Please refer to the attachment file for explicit comments.

Reviewer #4 (Remarks to the Author):

all comments were properly adressed

REVIEWERS' COMMENTS:

Reviewer #1 (Remarks to the Author):

I am happy with the changes the authors provided to their MS - I think the paper has improved in impact.

Just one more thing:

Line 255: impose a cost ON the performance

We made this correction. Thank you for improving our manuscript.

Reviewer #4 (Remarks to the Author):

all comments were properly addressed

Thank you for improving our manuscript.

Reviewer #3 (Remarks to the Author):

I agree with the other two reviewers that the ms is well prepared. However, it could be further improved.

Thank you for improving our manuscript. We have addressed each comment below.

Q: The answer does not sound good. It is weird to say that 26 °C was chosen because it is the median of the three treatments. Please give us a reasonable explanation. I believe that those three temperatures were chosen for their ecological importance. I doubt if it is reasonable to define 17 °C as a normal temperature and 26 °C as an increased temperature. Was 26 °C used because it is an optimal temperature for the development of the ladybeetle?

L88-95: 17 °C is defined as a normal temperature, but adults were maintained at 26 °C, an increased temperature. Why not maintain them at 17 °C? It seems more reasonable to maintain at the normal temperature.

Although 17 °C is a normal temperature throughout the active season in the field in the Czech Republic, temperatures used for rearing adult ladybirds, facilitating mating, and oviposition in the lab are often higher (e.g., 20°C and 25°C in Knapp and Nedvěd 2013; 26 °C in Knapp and Řeřicha 2020). We made the experimental decision in this study to use the increased temperature (26°C) as our rearing temperature and post-acclimation temperature because it is the median of the three treatments.

Our decision to keep beetles at 26 °C was a mixture of practical aspects and one of them was the fact that under laboratory conditions 26 °C seems to be closer to optimal rearing temperature than 17 °C. We decided to omit any detailed explanations from the manuscript text as the experimental setting is already given (impossible to change) at the publication stage.

Q: Sorry, what is the minimum value?

L134: A maximum of 12 individuals were used, what is the maximum value? Please directly show it in the main text.

We have changed the sentence to be clear. We used maximally 12 individuals at a time to measure HKDt (per batch). Line 133: “To measure HKDt, sets of maximally 12 individuals were placed in clean Petri dishes (without food and water) in a climatic chamber set to 42°C.”

Sorry for the confusion. We now understand the comment and our response was not helpful. We mean to say that 12 individuals were watched at a time (never more). We have edited the text to reflect this.

Line 133: To measure HKDt, sets of 12 individuals (in few cases less) were placed in clean Petri dishes (without food and water) in a climatic chamber set to 42°C.

Q: If the following explanation holds, then there was not a lasting effect on adult immune function because there was a significant day effect. Right?

L263: There were no lasting effects on heat tolerance. Please check it.

From the GLMM, there was an overall lasting effect on heat tolerance since we did not find a statistically significant effect of day on heat knock down time (HKDt). Below is the specific section from the results.

Line 218: “Compared to cold tolerance (CCRt), HKDt values were similar for adult ladybirds measured 1 and 7 days after eclosion (Figure 2; Table S2; Day main effect: $p = 0.30$; Temperature*Day interaction effect $p = 0.46$).” We think this justifies the line in question (Line 263 in the discussion).

We agree, however, we are still convinced that our statement (in the manuscript) works well. Lasting effects doesn't only mean that values are the same at day 1 and day 7 (the case of HKDt), but also the case when values change from D1 to D7 (significant day effect) but differences between treatments are still significant (the case of immune function). In contrast, CCRt is the case when significant differences disappeared = there is no lasting effect of temperature on CCRt.

Q: I still doubt that there is a significant difference between 26 °C and 35 °C treatments. Could you please show the results of the model check from the check_model function?

Figure 3: Starvation resistance: I do not think there is significant difference between 26 and 35. Because there is much overlap between the values, even if they are shown in mean \pm SE. Please check it. In addition, I want to know how you check the validation of model fitting.

We have double checked the result and can confirm that based on our statistical analyses that those values are significantly different. We have added details to the text describing how we validated our models. Line 202: “The performance of each model was visually inspected using the check_model function from the performance package.”

We have provided the output from the check from the check_model function as well as the output from the posthoc test used to examine potential differences between temperature treatments.

contrast	estimate	SE	df	t.ratio	p.value
T17 - T26	-0.383	0.501	119	-0.765	0.725
T17 - T35	-2.516	0.5	119	-5.036	<.0001
T26 - T35	-2.133	0.524	119	-4.073	0.0002

Q: For Figure 4, I prefer to add error bars to the plot. The plot may then be further improved by using smaller points. You can also try to “jitter” the point between treatments (staggered points for the same day between treatments, for example, set $x+0.1$ for 17 °C, x for 26 °C, and $x-0.1$ for 35 °C), in that way it may be easier to distinguish the error bars.

Figure 4: The 95% confidence intervals for each point are missing.

For clarity and to remove clutter, we decided to remove the error bars for this figure. Attached is the figure with the error bars.

We have made a final version of this figure with these improvements.

Table S2: I would like to reorganize it like this:

Model number	Response variable	Experimental variables	df	X2	p
1	Chill coma recovery time	Live mass			
		Sex			

		Temperature			
		Day			
		Temperature *day			
2	Heat knock down time	Dry mass			
		Sex			
		Temperature			
		Day			
		Temperature*day			

We have made this change and the table is similar to the example you provided. Thank you for the improvement.

Q: L122 and L203: Replace ‘time until first clutch’ with ‘preoviposition period’.

We have made this change.

Q: As suggested by reviewer 1, it is better to indicate the pupal development time under three acclimation temperatures.

We addressed reviewer #1’s comment by demonstrating that the main concern of development time on adult size was not a factor in our results.